# A Review on the Current Treatment Paradigm in High-Risk Prostate Cancer

**DOI:** 10.3390/cancers13174257

**Published:** 2021-08-24

**Authors:** Laura Burgess, Soumyajit Roy, Scott Morgan, Shawn Malone

**Affiliations:** 1Division of Radiation Oncology, Department of Radiology, University of Ottawa, Ottawa, ON K1H 8L6, Canada; smorgan@toh.ca; 2Radiation Medicine Program, The Ottawa Hospital Cancer Centre, Ottawa, ON K1H 8L6, Canada; 3Department of Radiation Oncology, Rush University Medical Center, Chicago, IL 60605, USA; soumyajit_roy@rush.edu

**Keywords:** high-risk prostate cancer, radiotherapy, molecular imaging, targeted therapy

## Abstract

**Simple Summary:**

Patients with high-risk prostate cancer are usually treated with combination of radiotherapy and androgen deprivation therapy. However, there has been long strides of advancements in the domain of radiotherapy and systemic therapy in the last decade. Similarly, there has been significant improvement in the surgical sphere. Additionally, significant improvements in the genomic classifiers and imaging modalities have widened the scope of improved risk stratification and personalization of treatment in this patient population. In this study we have reviewed the modern paradigm of management of patients with high-risk prostate cancer in light of the emerging evidence.

**Abstract:**

High-risk prostate cancer is traditionally treated with a combination of radiotherapy (RT) and androgen deprivation therapy (ADT). However, recent advancements in systemic treatment and radiotherapy have widened the spectrum of treatment for this patient population. Use of image guidance and intensity modulation, as well as the incorporation of brachytherapy, has led to safe radiotherapy dose escalation with reduced risk of recurrence. Clinical trials have helped define the role of pelvic nodal radiotherapy, the role of stereotactic ablative radiotherapy, and the optimal duration and sequencing of ADT in combination with radiotherapy. Emerging evidence has redefined the role of surgery in this cohort. Contemporary clinical trials have identified new systemic therapy options in high-risk prostate cancer. Finally, new imaging modalities including multi-parametric MRI and molecular imaging and genomic classifiers have ushered a new era in patient selection, risk stratification, and treatment tailoring.

## 1. Introduction

Approximately 17–31% of newly diagnosed men with prostate cancer present with non-metastatic high-risk localized or locally advanced disease [1]. Compared to low-risk prostate cancer, these patients have a more than threefold higher risk of death from cancer-related causes [2,3]. High-risk prostate cancer is characterized by D’Amico criteria as any of the following—clinical stage T3a or higher, Gleason score 8 to 10, and pre-treatment prostate-specific antigen (PSA) > 20 ng/mL [2]. The National Comprehensive Cancer Network (NCCN) categorizes these patients into high- and very high risk groups, with the latter consisting of patients with T3b or T4 tumors, primary Gleason score of 5, at least 5 cores with grade group 4 or 5, or at least two high-risk features [4]. While these stratification systems provide useful prognostic information that is used in the clinical setting, they do not capture all characteristics that determine biological aggressiveness of tumors or the genetic factors that may play a role [5]. Additionally, there remains significant heterogeneity in overall outcomes among high-risk prostate cancer patients. Molecular biomarkers have the potential to improve the risk stratification algorithm. This is reflected in the most recent NCCN 2020 guidelines that incorporate the use of biomarkers in more aggressive patient populations with longer life expectancy. In this population, curative-intent treatment such as combination of long-term androgen deprivation therapy (ADT) with radiotherapy (RT) confers a survival benefit [6]; however, there remains a number of unanswered questions. These include equipoise on radiotherapy target volume, especially considering interaction of radiotherapy target volume with sequencing of ADT and radiotherapy, the ideal combination of newer systemic agents with the existing combination of radiotherapy and ADT, the use of ultra-hypofractionated radiotherapy, and the extent of pelvic nodal dissection, among others. Furthermore, there has been significant advancement in the field of imaging modalities and genomics-based prediction for prostate cancer. However, there remains lack of clarity on how to integrate these findings to personalize treatment in this patient population. Additionally, some of the recent studies have redefined the role and extent of surgery in these patients. Herein, we review the management of high-risk prostate cancer in light of the existing literature.

## 2. Materials and Methods

For the purposes of this narrative review, we subdivided studies into advances in radiotherapy, new imaging modalities, advances in systemic therapy, and the evolving role of surgery as part of multi-modal therapy in high-risk prostate cancer. Randomized trials, prospective non-randomized trials, retrospective reviews, and pooled analyses were included. Endpoints of interest included biochemical progression-free survival, overall survival, prostate cancer-specific survival, metastasis-free survival, and treatment-related toxicity.

## 3. Advances in Radiotherapy

The efficacy of RT in the treatment of localized prostate cancer is correlated to dose [7]. Dose-escalated RT is now standard of care in prostate cancer, either through external beam radiotherapy (EBRT) or brachytherapy boost [8,9,10,11,12]. Safe dose escalation has been made possible by technological advances, allowing for higher precision in dose delivery and sparing of organs at risk.

Among these technological advances, image-guided radiotherapy (IGRT) is pivotal. This allows for visualization, whether direct or indirect, of the prostate at various times throughout treatment, allowing for correction for displacement of the prostate and improved disease control, while minimizing toxicity [13,14]. A randomized phase III clinical trial including 470 men demonstrated that when compared to weekly IGRT, daily IGRT resulted in significantly improved biochemical and clinical progression-free interval, as well as improved acute and late rectal toxicity [15].

It is thought that the major benefit of dose escalation is in improved local tumor control. One study of 143 men with high-risk prostate cancer showed that dose escalation decreases the rate of positive post-RT biopsies; 51% of men had positive prostate biopsies following RT with doses less than or equal to 70.2 Gy, while only 15% had positive prostate biopsies following RT with doses greater than 81 Gy [16].

ASCENDE-RT randomized men with intermediate- and high-risk prostate cancer into two different dose-escalation strategies; dose-escalated EBRT and dose-escalation using an LDR brachytherapy boost [8]. They found that after median follow-up of 6.5 years, compared to 78 Gy EBRT, men treated with LDR brachytherapy boost were twice as likely to be free of biochemical failure (hazard ratio (HR): 2.04; 95% confidence intervals: 1.25–3.33), with 7-year biochemical progression-free survival 86% in brachytherapy arm and 75% in the dose-escalated EBRT arm [8]. Acute and late GU morbidity was three times higher in the brachytherapy arm, and there was a trend towards increased GI morbidity, although not statistically significant [17]. Updated analysis demonstrated long-term durable biochemical control; 10-year and 15-year biochemical progression-free survival of 85% and 80% in the brachytherapy arm and 67% and 53% in the external beam arms, respectively (HR: 2.0, 95% confidence intervals (CI): 1.3–3.1) [18]. This durable improvement in biochemical control did not translate into an overall survival benefit, likely due to greater use of salvage ADT at relapse in the EBRT arm (24% vs. 13%). 

The addition of a brachytherapy boost may offer some benefit in men with high-risk disease and adverse prognostic factors. Findings from TROG 03.04 RADAR raised a possibility of such benefit. This study evaluated dose escalation, together with the timing of ADT in men with locally advanced prostate cancer. Patients were treated with 66 Gy, 70 Gy, or 74 Gy with EBRT or 46 Gy EBRT plus an HDR brachytherapy boost. Independent of ADT duration, HDR brachytherapy boost significantly reduced 10-year distant progression [19]. A retrospective review of 131 patients with very high risk features (at least one of Gleason score 10, Gleason score 8 or 9 with PSA > 20 ng/mL, clinical T3, >50% positive cores, PSA > 40 ng/mL) treated with EBRT, LDR brachytherapy boost, and ADT had 12-year cancer-specific survival of 87%, biochemical progression-free survival of 87%, and overall survival of 61% [20]. Overall, these findings suggest that dose escalation using brachytherapy boost should be considered as an option to optimize outcomes in a very high-risk setting. 

Another approach is to offer dose escalation specifically to the dominant lesion(s) within the prostate, rather than to the entire prostate. Development and incorporation of newer imaging modalities such as multiparametric MRI (mpMRI) and positron emission tomography (PET) facilitates such an approach. FLAME is a phase III multicentric trial that investigated the role of focal boosts, with a boost to the macroscopically visible tumor on mpMRI up to 95 Gy [21]. The dose-escalated arm had significantly improved five-year biochemical disease-free survival (HR: 0.45, 95% CI: 0.29–0.71), without increased GU or GI toxicity. Organ at risk constraints were prioritized over focal boost dose. Acknowledging the limitation of a short follow-up duration, the authors did not find any benefit with respect to prostate cancer-specific survival (HR: 0.69, 95% CI: 0.27–1.79) or overall survival (HR: 1.26, 95% CI: 0.83–1.92). There are also studies that used PET to intensify radiotherapy to visible tumor [22] and others involving the use of stereotactic body radiotherapy (SBRT) and hypofractionation [23]. This treatment approach still needs further validation but may help to safely improve local tumor control. Despite the dose escalation studies showing improved failure-free survival, there remains a lack of overall survival benefit with such approach. Distant failure from pre-existing micrometastatic disease, which is unlikely to be eradicated through local treatment intensification, remains a plausible mechanism explaining the failure of this strategy to attain any overall survival benefit. 

While targeted SBRT to focal lesions is still under investigation, there is evidence that ultra-hypofractionated SBRT to the whole prostate is safe and effective in low- and intermediate-risk prostate cancer [24]. Studies have also explored its safety and efficacy in high-risk settings. A meta-analysis evaluated the role of SBRT in the treatment of localized prostate cancer. A total of 38 prospective studies with 6116 patients were included [25]. Of these, 38% of studies included patients with high-risk prostate cancer. They found that SBRT provided excellent biochemical control, with 5-year biochemical progression-free survival of 93.7% (95% CI: 91.4–95.5%). The study also demonstrated that higher dose of SBRT was associated with greater biochemical control (*p*-value 0.018), however at the cost of worse late grade 3 or worse GU toxicity (*p*-value 0.014) [25]. HYPO-PROST is a randomized trial that compared ultra-hypofractionated RT boost (15 Gy in two fractions) to conventionally fractionated RT boost (30 Gy in 15 fractions), in addition to 46 Gy to the pelvis in men with high-risk prostate cancer. There was no difference in 5-year biochemical relapse-free survival (HR: 1.35, 95% CI: 0.72–2.53), metastasis-free survival (HR: 1.3, 95% CI: 0.62–3.27), or overall survival (HR: 2.01, 95% CI: 0.80–5.05) between arms, and no significant differences was noted in grade 2 or worse late GI or GU toxicities [26]. Similarly, HYPO-RT-PC is a randomized phase III non-inferiority trial that included 1180 men with intermediate- (89%) and high-risk (11%) prostate cancer [27]. Patients were randomized to ultra-hypofractionation (42.7 Gy in seven fractions) or conventional fractionation (78.0 Gy in 39 fractions). There was no significant difference in failure-free survival at 5 years (HR: 1.002, 95% CI: 0.758–1.325). There was a mild increase in urinary toxicity at 1 year in the ultra-hypofractionated group (6% grade 2 or worse in ultra-hypofractionated group vs. 2%, *p*-value 0.0037), but there was no difference in GI toxicity.

Elective nodal irradiation in high-risk prostate cancer is somewhat controversial. GETUG-01 and RTOG 9413 did not demonstrate any benefit to biochemical progression-free survival or overall survival with the addition of pelvic nodal radiotherapy [28,29]. RTOG 0534 SPPORT trial investigated the role of prostate bed radiotherapy alone, the addition of short-term ADT, and the addition of both short-term ADT and elective nodal radiation in the post-operative salvage setting. Interim analysis revealed 5-year biochemical progression-free survival of 71%, 83%, and 89%, respectively, in the three arms [30]. This was suggestive of the possible benefit of nodal radiotherapy and has been further supported by a recently published phase III randomized trial POP-RT. This randomized phase III study compared whole pelvis radiation to prostate-only radiation in high-risk node-negative prostate cancer. Overall, 228 patients were included. Importantly, approximately 80% of patients enrolled underwent PSMA PET-CT for initial staging. At a median follow-up of 68 months, whole-pelvis radiotherapy improved 5-year biochemical progression-free survival (HR: 0.23, 95% CI: 0.10–0.52), 5-year disease-free survival (HR: 0.40, 95% CI: 0.22–0.73), and metastasis-free survival (HR: 0.35, 95% CI: 0.15–0.82) compared to prostate-only radiotherapy, but did not impact overall survival (HR: 0.92, 95% CI: 0.41–2.05) [31]. However, the follow-up in these two studies is still premature in terms of detecting any overall survival difference. With time, it will become clearer if the addition of pelvic nodal radiotherapy improves overall survival in high-risk prostate cancer. Furthermore, ongoing studies such as RTOG 0924 will clarify if there is a survival advantage with use of elective nodal irradiation in high-risk prostate cancer.

## 4. Systemic Therapy for High-Risk Prostate Cancer

Two large randomized studies have demonstrated superior overall survival with long-term ADT relative to short-term ADT when combined with radiotherapy [32,33]; however, the optimal duration of ADT remains largely undefined. This lack of clarity is mostly secondary to the fact that many different durations have been studied across different studies. RTOG 9202 compared 4 months of ADT to 28 months of ADT. After a median follow up of 19.6 years, long-term ADT improved the rate of distant metastases, disease-specific survival, and overall survival [33]. On the other hand, EORTC 22961 compared 6 months and 36 months of ADT in conjunction with EBRT. After a median follow up of 6.4 years, 6-months of ADT was associated with a 42% increased risk of overall mortality (HR: 1.42 with post hoc 95% CI: 1.09–1.85) compared to 36 months of ADT [32]. These two trials set the standard of care for ADT duration to 24 to 36 months, which is also endorsed by the NCCN guidelines [4]. The Prostate Cancer Study IV compared outcomes in men with node-negative high-risk prostate cancer treated with radiation with either 18 moths or 36 months of ADT [34]. The HR of death was 1.15 with 95%CI of 0.85 to 1.56, and the authors concluded that ADT duration can safely be reduced to 18 months. Nonetheless, there is significant controversy around this conclusion. This controversy stems from the very wide confidence interval and the upper bound of the HR of 1.56, greater than the commonly cited upper bound for non-inferiority trials of 1.25 [35]. The results are also confounded by poor adherence to the intended duration of ADT in the 36-month arm.

Sequencing of systemic therapy with local therapy has been shown to influence the overall outcome in several types of malignancies including prostate cancer. Two randomized trials (OTT-0101 and RTOG 9413) evaluated the sequencing of short-term ADT with prostate-directed radiotherapy. Findings from both these studies raised a possibility of superior biochemical relapse-free survival or progression-free survival with adjuvant ADT with prostate directed radiotherapy, respectively [29,36]. An individual patient data-based meta-analysis of these two studies demonstrated that the sequencing of ADT with prostate-directed RT has significant impact on long-term oncologic outcomes in patients with localized prostate cancer [37]. Up to a 15-year truncation point in follow-up, restricted mean survival time for progression was significantly higher with adjuvant ADT with an absolute difference of 10.8 months (95% CI: 2.7–18.8), favoring the adjuvant approach. Adjuvant ADT led to significant reduction in the risk of relative incidence of distant metastasis (sub-distribution HR: 1.40, 95% CI: 1.00–1.95) and non-significant reduction in the incidence of cancer-specific mortality (sub-distribution HR: 1.29, 95% CI: 0.95–1.75). There was no interaction of NCCN risk group with the benefits obtained with adjuvant strategy. Moreover, adjuvant short-term ADT did not result in any increase in the risk of clinician-reported toxicity or any deterioration of patient-reported outcomes [38]. Although an independent trial would be the best way to investigate the validity of these findings with respect to use of long-term ADT, such a trial would require a large patient population with long-term follow-up. Until then, one should consider a concurrent and adjuvant approach while combining long-term ADT with radiotherapy in high-risk prostate cancer. 

The utility of combining ADT with radiotherapy has often been questioned, especially in the setting of dose escalation. In fact, NCCN guidelines for both high- and very high risk prostate cancer include ADT as an option, rather than necessity, when EBRT is combined with low dose rate (LDR) brachytherapy boost [4]; however, a recent network meta-analysis suggests that EBRT and ADT may result in superior overall survival compared to patients treated with EBRT and brachytherapy boost (HR 0.68, 95% CI: 0.52–0.89) [39]. In the DART01/05 GICOR study, where about 50% patients belonged to the high-risk category, 28 months of long-term ADT combined with 78 Gy of external beam radiotherapy (EBRT) was associated with significantly superior overall and metastasis-free survival compared to 4 months of ADT with same dose of EBRT [40]. In the TROG 03.04 RADAR trial, where approximately two-thirds of patients belonged to the high-risk group, it was found that 18 months of ADT reduced distant progression compared to 6 months of ADT, regardless of radiotherapy dose [19]. Similarly, in the EORTC 22991 trial, 6 months of adjuvant ADT significantly improved biochemical disease-free survival and clinical progression-free survival across all radiation dose levels (70 Gy vs. 74 Gy vs. 78 Gy) [41]. All these findings, taken together, confirm that dose-escalation and ADT synergize with each other by enhancing local and distant control, respectively.

Several newer systemic agents have shown superior outcomes when used in the treatment of metastatic castrate-sensitive and non-metastatic castrate-resistant prostate cancer. These include docetaxel, abiraterone, apalutamide, enzalutamide, and darolutamide [42,43,44,45,46,47,48,49,50,51]. Some of these agents could show promising results when combined with RT and ADT in high-risk or locally advanced prostate cancer. 

RTOG 0521 studied the role of adjuvant docetaxel in combination with RT and long-term ADT in patients with high-risk non-metastatic prostate cancer. Of the 563 evaluable patients, 53% had Gleason score 9 to 10 and 27% had clinical T3 or T4 disease. After a median follow up of 5.7 years, four-year overall survival was 89% in the RT + ADT arm and 93% in the RT + ADT and adjuvant docetaxel arm (HR: 0.69, 90% CI: 0.49–0.97) [52]. The number needed to treat to prevent one death is 25 patients. Additionally, adverse events related to treatment were roughly twice as common with the addition of docetaxel. GETUG-12 also evaluated the role of docetaxel in the high-risk setting, randomly assigning 413 men with high-risk prostate cancer to long-term ADT with and without four cycles of docetaxel and estramustine, following staging pelvic lymph node dissections. Relative to ADT alone, 12-year relapse-free survival was significantly improved with docetaxel-based chemotherapy (36.3% vs. 49.4%, HR: 0.75, 95% CI: 0.55–0.93). However, there was no significant difference in the rates of metastasis-free survival and cancer-specific survival between the two arms [53]. 

STAMPEDE is multi-arm, multi-stage design of randomized controlled trials that has investigated the role of both docetaxel and abiraterone among men with high-risk, locally advanced, and metastatic prostate cancer. Approximately 20% of patients in the docetaxel portion of the study had high-risk localized and locally advanced prostate cancer. The addition of docetaxel to standard of care treatment improved overall survival (HR: 0.81, 95% CI: 0.69–0.95), and this benefit was independent of the metastatic stage of the cancer [54]. A meta-analysis of trials comparing standard of care with or without docetaxel in the high-risk localized setting demonstrated significant improvement in failure-free survival with addition of docetaxel (HR: 0·70, 95% CI: 0·61–0·81). Nonetheless, no overall survival benefit was obtained with chemo–hormonal combination (HR: 0.87, 95% CI: 0.69–1.09) [55]. On the basis of these findings, docetaxel remains an option for high-risk patient population after appropriate discussion on benefits and morbidities with the patient. 

Approximately one-quarter of patients in the abiraterone arm of the STAMPEDE study had high-risk, localized prostate cancer. For the overall study population, abiraterone plus prednisolone combined with ADT was shown to have significantly higher failure-free survival and overall survival (HR: 0.63, 95% CI: 0.52–0.76; HR 0.61 in non-metastatic arm) compared to ADT alone, and there was no heterogeneity of treatment effect across metastatic and non-metastatic subgroups [56]. In the subgroup of patients with non-metastatic disease, the addition of abiraterone had profound benefit in the intermediate endpoint of failure-free survival (HR: 0.21, 95% CI: 0.15–0.31). Matured overall survival results for this subgroup are awaited. A phase II single-arm prospective trial of 37 men with unfavorable intermediate or high-risk localized prostate cancer evaluated the role of 6 months of ADT with abiraterone plus prednisone, in combination with RT [57]. After median follow-up of 46 months, the study showed a promising PFS outcome. The reported 3-year PFS was 96% (95% CI: 76–99). 

Other systemic agents that are under investigation in high-risk localized prostate cancer setting include apalutamide [58], enzalutamide [59], darolutamide [60], immunotherapy [61], and poly(adenosine diphosphate-ribose) polymerase (PARP) inhibitors [62].

Two phase III studies are investigating the addition of apalutamide (ATLAS, NCT02531516 with primary endpoint of metastasis-free survival) and enzalutamide (ENZARAD, NCT02446444 with primary endpoint of overall survival) combined with ADT for patients with high-risk prostate cancer undergoing primary radiation therapy. Both studies have completed accrual. These studies will provide further clarity on the utility of these agents in high-risk prostate cancer. 

KEYNOTE-1999 is a phase II study investigated the anti-cancer activity of pembrolizumab, an anti-PD-1 monoclonal antibody, in the treatment of men with metastatic castrate-resistant prostate cancer. Disease control rates varied between 9% and 22%, varying with PD-L1 expression [61]. While this study did not include high-risk localized patients, it nonetheless demonstrates that the exciting advances seen with the use of immunotherapy in other malignancies may be possible in prostate cancer. In a similar population of men with metastatic castrate-resistant prostate cancer, a phase III trial evaluated the role of the PARP inhibitor olaparib in men whose disease had progressed with a new hormonal agent. PARP inhibitors are known to impact patients with loss-of-function gene alterations, including homologous recombination repair, as such all men had alterations in prespecified genes involved in homologous recombination repair BRCA1, BRCA2, or ATM in cohort A, and in cohort B, alterations in 12 DNA damage repair genes were allowed. Olaparib significantly improved rates of progression or death relative to controls (HR: 0.34, 95% CI: 0.5–0.47, *p* < 0.001) [62]. There is also some suggestion that there may be benefit of adding PARP inhibitiors to patients with earlier-stage disease, particularly high-risk prostate cancer. Asim et al. demonstrated that ADT can functionally impair homologous recombination prior to castration resistance; thus, upfront use of PARP inhibitors together with ADT may be beneficial in high-risk prostate cancer [63]. We know that certain populations with localized prostate cancer do poorly; for instance, patients with germline BRCA mutations have worse outcomes than non-carriers [64], and they may especially benefit from upfront use of PARP inhibitors. 

These studies also highlight the role of personalization in systemic therapy, with expression of specific genes playing a role in response to treatment. A number of genes have been identified to play a critical role in prostate cancer [65], and we will likely see incorporation of new systemic therapy agents in high-risk prostate cancer—often guided by molecular markers—in the years to come. Appendix A summarizes the advantage and disadvantages of different treatment strategies in high-risk prostate cancer.

## 5. Importance of New Imaging Modalities and Molecular Imaging

Conventional clinical staging with digital rectal exam, PSA, random and systematic biopsies, with or without CT, pelvic lymphadenectomy, and bone scans have modest accuracy and the ability to define both location and burden of disease [66]. In fact, it is estimated that conventional staging underestimates the location and burden of disease by 20–30% [67], which can lead to undertreatment and subsequent relapse of disease. In the high-risk setting, the window for cure is correlated to the accurate detection of the extent of clinically significant prostate cancer [68].

Disease identification has been improved with the use of mpMRI and positron emission tomography (PET). MpMRI, with T2-weighted (T2W) images, diffusion-weighted imaging (DWI), and dynamic contrast-enhanced images, are state-of-the-art imaging modalities to assess the local extent of prostate cancer [66,68]. The use of mpMRI increased the rate of detection of clinically significant cancer by 30% compared to the standard 12-core biopsy [67]. Specifically in the context of high-risk disease, T2W images have close to 100% sensitivity and specificity for identifying seminal vesicle invasion, and sensitivity as high as 75–90% for identifying extraprostatic extension [69]. 

Prostate-specific membrane antigen (PSMA) is overexpressed in more aggressive prostate cancer cells, making it a popular target for PET imaging. ProPSMA is a phase III randomized trial that compared staging with conventional imaging (CT and bone scan), with ^68^Ga-PSMA PET in men with high-risk prostate cancer [70]. Of the 295 men with follow up, 87 had pelvic nodal or distant metastatic disease. PSMA PET had 27% (95% CI: 23–31%) greater accuracy than conventional imaging at detecting this, with conventional imaging having a lower specificity (91%, 95% CI: 85–97%, vs. 98%, 95% CI: 95–100%) and lower sensitivity (38%, 95% CI: 24–52%,vs. 85%, 95% CI: 74–96%) than PSMA PET [70]. A study of 53 patients who underwent mpMRI and ^68^Ga-PSMA PET prior to radical prostatectomy showed that the combination of PET and mpMRI had a 98% sensitivity, whereas the sensitivities of detection for mpMRI and PET alone were 66% and 92%, respectively [71]. Similarly, a study of 34 patients’ post-radical prostatectomy who received either adjuvant or salvage radiotherapy with subsequent biochemical failure and no evidence of recurrence on conventional imaging evaluated the role of mpMRI and ^18^F-PSMA PET in staging [72]. A total of 32 of the 34 patients were noted to have ^18^F-PSMA PET avid lesions, with 17 (53.1%) having metastatic disease, 8 (25.0%) having locoregional recurrences, and 7 (21.9%) with local failure within the prostate bed. Of these, six lesions were in-field recurrences covered by the 100% isodose lines [72]. 

Studies have also evaluated the utility of advanced imaging techniques in redefining the role of radiation therapy in localized or locally advanced prostate cancer. We have already discussed the FLAME study and its findings previously. In addition to this, EMPIRE-1 is a single-center phase II/III trial that evaluated the utility of PET-directed radiotherapy in post-prostatectomy patients. In this study, 165 men with detectable PSA after prostatectomy and negative conventional imaging were randomized to radiotherapy directed by conventional imaging alone or conventional imaging and ^18^F-fluciclovine-PET/CT [73]. The use of ^18^F-fluciclovine-PET/CT led to a significant improvement in 3-year event-free survival (3-year event-free survival of 75.5% vs. 63%; HR for conventional imaging group: 2.05, 95% CI: 1.06–3.93) [73]. Menard et al. conducted a phase II randomized controlled trial using PSMA PET/CT to intensify RT, with an objective of improving failure-free survival outcome [22]. A total of 136 patients with either high-risk prostate cancer or biochemical failure after radical prostatectomy were enrolled. New lesions were detected in almost half the patients, guiding intensification of radiotherapy. In 75% of the cases, the sites of intensification were located outside of the initial planned radiotherapy fields. Findings of this study have encouraged the authors to evaluate this hypothesis in a randomized phase III trial [22]. Similar trials are underway assessing mpMRI-guided boosts using hypofractionation and ablative radiotherapy. These include hypo-FLAME, DELINEATE [23], SPARC, and 5STAR-PC, among others. 

These studies highlight the role of new imaging modalities and their potential role to guide personalized treatment for patients. 

## 6. Optimizing Local Therapy: Surgery as Part of a Multi-Modal Treatment

Traditionally, radical prostatectomy (RP) was predominantly used in low- and intermediate-risk localized prostate cancer. Some oncologists discouraged surgery in the high-risk setting because of low probability of disease control because of the risk of systemic micro-metastases [74,75,76,77]. In select high-risk patients, surgery has the potential to reduce or eliminate the burden of cancer and therefore minimize subsequent relapse and the need for subsequent treatments [78].

Overall, within the high-risk population, retrospective reviews suggest similar outcomes in patients treated with RP or RT. Boorjian et al. found no difference in 10-year prostate cancer-specific survival in men with high-risk prostate cancer treated with RP or RT and ADT [79], although there were more T3/4 tumors and Gleason score 8–10 patients in the RT and ADT group. A retrospective review comparing RP and RT in patients with Gleason score 9 and 10 prostate cancer found similar prostate cancer-specific survival between RP and RT [80]. However, those treated with dose-escalated RT had improved metastasis-free survival at 5 years. Another retrospective cohort analysis compared RP to RT and ADT in men with high-risk disease. Despite the fact that the RT arm was more likely to have higher tumor stage and Gleason 9 or 10 disease, there was no difference in prostate cancer-related deaths between the two groups [81]. In locally advanced prostate cancer, analysis of 1093 men in the Surveillance, Epidemiology and End Results database found greater survival in men treated with RP compared to those treated with RT, ADT, or a combination of both [82]. The findings of these studies are biased secondary to non-randomized treatment allocation, presence of unmeasured confounders, and missing data. However, RP is a viable treatment option for high-risk prostate cancer, particularly when combined with other modalities. A study of 1914 men with locally advanced prostate cancer based on the National Cancer Database demonstrated that local therapy with RP, RT, or a combination of both (18.6%) had improved 5-year overall survival compared to systemic therapy [83]. CALGB 90203 is a phase III study that investigated the role of neoadjuvant chemohormonal therapy (ADT and docetaxel) with radical prostatectomy in high-risk prostate cancer [84]. This demonstrated that the use of neoadjuvant chemohormonal therapy was associated with lower pathologic T-stage, likelihood of seminal vesicle invasion, likelihood of positive pelvic lymph nodes, and likelihood of positive surgical margins, compared to RP alone [84].

In patients undergoing RP, pelvic lymph node dissection (PLND) allows for lymph node staging and removal of any microscopic or macroscopic nodal disease. The therapeutic value of PLND remains controversial. There are two recent prospective single-center trials that randomized patients to limited PLND of obturator nodes or extended PLND of obturator, external iliac, internal iliac, common iliac, and presacral nodes. In one of them, 300 men with intermediate- and high-risk prostate cancer were included. There was no significant difference in biochemical recurrence-free survival (HR: 0.91, 95% CI: 0.63–1.32), metastasis-free survival, or cancer-specific survival [85]. A larger trial with 1440 patients randomized patients to limited PLND of external iliac nodes or extended PLND of external iliac, obturator, and hypogastric nodes. After 3.1 years of median follow-up, there was no significant difference in biochemical recurrence (HR: 1.04, 95% CI: 0.93–1.15), without any significant changes in morbidity. 

At this point, there is no randomized data comparing surgery and radiation in the high-risk setting. SPCG-15 will randomize patients to RT (EBRT alone or with a brachytherapy boost) and 18–30 months of ADT or RP with extended lymph node dissection with or without immediate or delayed post-op RT and ADT [86]. This will provide first prospective evidence of the role of surgery in high-risk patients, likely elaborating the potential benefit of RP in the multimodal setting. The use of MRI and PSMA PET may help select the patients appropriate for surgery or RT. We need to consider other individual patient prognostic factors, but patient preference should also play a key role in determining primary treatment modality [87,88].

## 7. Personalization of Therapy

In the last several years, there has been a movement in oncology towards personalized medicine, with the understanding that anti-cancer intervention is not one size fits all. This can be borne out in several ways in the high-risk localized prostate cancer setting. 

The incorporation of genomics into patient risk-stratification and treatment provides personalization of the treatment approach, and there are several commercially available panels available currently. It should be noted that most of these assays (Decipher, Prolaris, Oncotype DX, and Promark) were signed as prognostic, but not specifically predictive biomarker panels. The Decipher classification tool, a 22-gene genomic classifier, has been developed for men with prostate cancer [89]. Decipher genomic classification performs microarray analysis on the transcriptome to allow for further prognostication, guiding subsequent treatment. The algorithm is based on the expression of 22 RNA biomarkers important in cell proliferation and differentiation, androgen receptor signaling, cell motility, and immune response [90]. A recent systematic review examining the available evidence for its use in men with prostate cancer evaluated data from 30,407 patients, across 42 studies involving men with localized prostate cancer, post-prostatectomy, non-metastatic castrate-resistant cancer, and in the metastatic hormone-sensitive setting [91]. The study found that the Decipher genomic classification was prognostic for adverse pathology, biochemical failure, metastasis, cancer-specific survival, and overall survival. This applied to all disease states, but was strongest in the intermediate-risk and post-prostatectomy settings [91]. 

A number of other genomic tests show promise, including Prolaris, a molecular assay that uses 31 cell-cycle progression genes and 15 housekeeper genes to prognosticate prostate cancer-specific mortality through the generation of a score that reflects tumor cell proliferation [92]. Recently, it has been demonstrated that Prolaris provides a score that accurately predicts risk of progression to metastatic disease in men with intermediate- and high-risk prostate cancer [93].

Similarly, Oncotype Dx uses quantitative PCR tumor analysis of 12 genes associated with prostate cancer to provide a genomic prostate score (GPS). GPS was found to be associated significantly with increased odds of high-grade disease (OR: 2.3, 95% CI: 1.5–3.7) and non-organ confined disease (OR: 1.9; 95% CI: 1.3–3.0) and predicts for adverse pathology [94]. However, a recent prospective cohort trial, Canary PASS, did not find GPS score to be significantly associated with adverse pathology in men on active surveillance [95].

ExoDx is another personalized tool that aims to reduce unnecessary prostate biopsies through a novel urine exosome gene expression assay, together with PSA. It has been demonstrated to differentiate high-grade from low-grade prostate cancer in an initial validation study [96] and, more recently, in a two-phase prospective adaptive decision impact trial [97]. 

Lastly, ProMark is a gene profile assay that uses eight protein markers to provide a probability of detecting adverse pathologic factors in post-operative specimen through a biomarker score from zero to one, where biomarker score over 0.8 is associated with adverse pathology 76.9% of the time [98].

We will likely see increased utilization of such genomic approaches, providing greater tailoring of treatment to individual patients. NRG GU009 is a prospective trial evaluating the role of Decipher to tailor therapy (intensify or de-intensify therapy) in high-risk prostate cancer on the basis of genomic risk [99].

## 8. Conclusions

The approach to management of high-risk prostate cancer continues to evolve with emergence of new technology, development of new systemic therapy agents, incorporation of new imaging modalities, and a focus on individualized medicine. Studies have demonstrated that long-term ADT is beneficial in this patient population, as is dose intensification of radiation therapy. Advances in imaging and delivery techniques will allow for further dose intensification, specifically at sites of gross disease while minimizing dose to the non-involved areas, helping to minimize late toxicity. We may see the implementation of local therapy in conjunction with newer systemic therapy options. As more treatment approaches become available, there will be a move towards an individualized approach tailoring local and systemic therapy on the basis of genomics-based classifiers in addition to traditional tumor characteristics and patient factors.

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
