# Peer review of "A Review on the Current Treatment Paradigm in High-Risk Prostate Cancer"

_cancers, 2021, doi:10.3390/cancers13174257_

Round 1
Reviewer 1 Report
Brief Summary:
The aim of the study by Burgess et al., was to provide a literature review on the current status and recent advancements of treatments against high-risk prostate cancer. The review provides updates on the clinical trials involving updates in radiotherapy and systemic (Androgen deprivation therapy) ADT treatments specifically targeted for patient with high-risk prostate cancer. Moreover, the authors touch upon the advancements in imaging modalities, such as MRI and PET scans and how these approaches could be used to redefine the role of radiotherapy, with an excellent mention on SBRT, and chemotherapy in high-risk patient groups. In addition, they offer a brief retrospective review for the role of radical prostatectomies and incorporation of genomics into precision prostate cancer treatment. Overall, this a comprehensive review focusing on specific aspects of treating high-risk prostate cancer and could be of interest to the readers. However, there are some minor issues to be addressed before the manuscript is considered for publication.
Comments
- The title seems to be too broad and does not incorporate the scope of the paper. The manuscript lacks an in-depth review of the genomic determinants for high-risk prostate cancer and how these can be used to influence treatment options. This is a huge part of the “evolving treatment landscape”. It is understandable and fully acceptable that the authors chose to focus their review on other treatment aspects, but I would highly recommend adjusting the title to avoid misleading the audience. [minor]
- In the introduction, the authors should provide a more thorough description of what is high-risk prostate cancer and which patients fall into this category. For example, recent studies have shown that the expression of certain biomarkers render patients more at risk for developing aggressive disease (PMID: 33893149, 32708810, 32412803, 32545454). The study would benefit from a better definition of high-risk prostate cancer, beyond just the purely clinical indications. [minor]
- In the systemic therapy section on ADT, the authors may be missing some key genetic determinants for chemotherapy response in high-risk patients. If possible, they should include a very brief mention on these. [minor]
- Although there is a small section on personalized therapy in the manuscript, the authors appear to focus only on the Decipher study and not other recent advancements on precision medicine approaches, such as hereditary genomics, POLARIS, PROMARK, ONCOTYPE (see PMID: 33893149, 32708810, 32412803, 32545454). A brief mention would also benefit the study. [minor]
Author Response
Reviewer 1
The aim of the study by Burgess et al., was to provide a literature review on the current status and recent advancements of treatments against high-risk prostate cancer. The review provides updates on the clinical trials involving updates in radiotherapy and systemic (Androgen deprivation therapy) ADT treatments specifically targeted for patient with high-risk prostate cancer. Moreover, the authors touch upon the advancements in imaging modalities, such as MRI and PET scans and how these approaches could be used to redefine the role of radiotherapy, with an excellent mention on SBRT, and chemotherapy in high-risk patient groups. In addition, they offer a brief retrospective review for the role of radical prostatectomies and incorporation of genomics into precision prostate cancer treatment. Overall, this a comprehensive review focusing on specific aspects of treating high-risk prostate cancer and could be of interest to the readers. However, there are some minor issues to be addressed before the manuscript is considered for publication.
Comments
- The title seems to be too broad and does not incorporate the scope of the paper. The manuscript lacks an in-depth review of the genomic determinants for high-risk prostate cancer and how these can be used to influence treatment options. This is a huge part of the “evolving treatment landscape”. It is understandable and fully acceptable that the authors chose to focus their review on other treatment aspects, but I would highly recommend adjusting the title to avoid misleading the audience. [minor]
We agree that our original title was too broad and did not perfectly reflect the content of the manuscript and we have modified the title accordingly to more accurately reflect the manuscript.
- In the introduction, the authors should provide a more thorough description of what is high-risk prostate cancer and which patients fall into this category. For example, recent studies have shown that the expression of certain biomarkers render patients more at risk for developing aggressive disease (PMID: 33893149, 32708810, 32412803, 32545454). The study would benefit from a better definition of high-risk prostate cancer, beyond just the purely clinical indications. [minor]
We have now incorporated discussion about the shortcomings of current stratification systems and the use of biomarkers and genetic factors that play a critical role in defining the aggressiveness of prostate cancer. We feel that this strengthens the conversation overall and appreciate the recommendation.
- In the systemic therapy section on ADT, the authors may be missing some key genetic determinants for chemotherapy response in high-risk patients. If possible, they should include a very brief mention on these. [minor]
We agree that this strengthens the manuscript overall and have added conversation about genetic determinants of response to treatment and various systemic therapies taking advantages of these such as PARP inhibitors.
- Although there is a small section on personalized therapy in the manuscript, the authors appear to focus only on the Decipher study and not other recent advancements on precision medicine approaches, such as hereditary genomics, POLARIS, PROMARK, ONCOTYPE (see PMID: 33893149, 32708810, 32412803, 32545454). A brief mention would also benefit the study. [minor]
We appreciate this suggestion and have added more discussion about other personalized medicine approaches. We feel that this addition has significantly strengthened the review of personalized medicine in this manuscript
Reviewer 2 Report
The authors made a review of the new and emerging treatment modalities for high-risk prostate cancer. A few suggestions to increase the utility of the manuscript:
1) It appears that due to the authors' specialty, more emphasis was placed on radiation therapy options for localized or locally-advanced high-risk prostate cancer. It would help to modify the title accordingly, or include advances in the metastatic disease setting and other treatment modalities.
2) A table summarizing the advantages/disadvantages of the various currently used/proposed treatment modalities would be helpful.
3) Please also describe other genomic tests such as ExoDx, OncotypeDx etc. and their potential utility in the high-risk setting.
4) Please also discuss other personalized therapy options such as immunotherapies and PARP inhibitors in the high-risk setting.
Author Response
The authors made a review of the new and emerging treatment modalities for high-risk prostate cancer. A few suggestions to increase the utility of the manuscript:
1) It appears that due to the authors' specialty, more emphasis was placed on radiation therapy options for localized or locally-advanced high-risk prostate cancer. It would help to modify the title accordingly, or include advances in the metastatic disease setting and other treatment modalities.
On your suggestion, we have modified our title. We believe that the new title more accurately represents the content of this manuscript.
2) A table summarizing the advantages/disadvantages of the various currently used/proposed treatment modalities would be helpful.
We have now included a table summarizing treatment modalities, their advantages, disadvantages and the relevant evidence for these. We feel that this provides an effective summary of treatment modalities.
3) Please also describe other genomic tests such as ExoDx, OncotypeDx etc. and their potential utility in the high-risk setting.
We have expanded our discussion of various genomic tests in the personalized medicine section, we feel that this addition has better summarized the role of genomics in the management of high-risk prostate cancer.
4) Please also discuss other personalized therapy options such as immunotherapies and PARP inhibitors in the high-risk setting.
We have included a review of other systemic therapy options including immunotherapy and PARP inhibitors and we feel that this strengthens the manuscript; we appreciate the suggestion.